

# Germline mutation analyses of malignant ground glass opacity nodules in non-smoking lung adenocarcinoma patients

Wenjun Mao[1,*], Ruo Chen[1,*], Rongguo Lu[1], Shengfei Wang[1], Huizhu Song[2], Dan You[2], Feng Liu[1], Yijun He[1] and Mingfeng Zheng[1]

[1] Department of Cardiothoracic Surgery, The Affiliated Wuxi People's Hospital of Nanjing Medical University, Wuxi, Jiangsu, China

[2] Department of Pharmacy, The Affiliated Wuxi People's Hospital of Nanjing Medical University, Wuxi, Jiangsu, China

[*] These authors contributed equally to this work.

## ABSTRACT

**Background.** Germline mutations play an important role in the pathogenesis of lung cancer. Nonetheless, research on malignant ground glass opacity (GGO) nodules is limited.

**Methods.** A total of 13 participants with malignant GGO nodules were recruited in this study. Peripheral blood was used for exome sequencing, and germline mutations were analyzed using InterVar. The whole exome sequencing dataset was analyzed using a filtering strategy. KOBAS 3.0 was used to analyze KEGG pathway to further identify possible deleterious mutations.

**Results.** There were seven potentially deleterious germline mutations. NM_001184790:exon8: c.C1070T in *PARD3*, NM_001170721:exon4:c.C392T in *BCAR1* and NM_001127221:exon46: c.G6587A in *CACNA1A* were present in three cases each; rs756875895 frameshift in *MAX*, NM_005732: exon13:c.2165_2166insT in *RAD50* and NM_001142316:exon2:c.G203C in *LMO2*, were present in two cases each; one variant was present in *NOTCH3*.

**Conclusions.** Our results expand the germline mutation spectrum in malignant GGO nodules. Importantly, these findings will potentially help screen the high-risk population, guide their health management, and contribute to their clinical treatment and determination of prognosis.

Corresponding author
Mingfeng Zheng,
zhengmfmedical@126.com

## INTRODUCTION

Though therapeutic advances have been made using targeted therapy and immunotherapy, lung cancer continues to be the most common cause of cancer-related deaths worldwide (*Siegel, Miller & Jemal, 2015*). The majority of lung cancers are caused by somatic mutations that accumulate with age and germline mutations could explain a predisposition to cancer development.

Lung cancer is a complex disease that is mainly attributed to smoking (*Hung et al., 2008*). However, over 10% of lung cancer patients are non-smokers (*Subramanian & Govindan,*

*2007*). The development of lung cancer in never-smokers is associated with several potential risk factors, including environmental pollution and genetic predisposition (*Malhotra et al., 2016*). Germline mutations in lung cancer have been studied to some extent (*Ikeda et al., 2014*; *Liu et al., 2016*; *Shukuya et al., 2018*; *Zhang et al., 2017*), including some in familial settings (*Kanwal et al., 2018*; *Tomoshige et al., 2015*). There are also studies on non-smoking lung cancer cohorts (*Donner et al., 2018*; *Renieri et al., 2014*). Nonetheless, studies on germline mutations in lung cancer patients fall far short when compared to those on somatic mutations. There is a need to study germline mutations in lung cancer since they are related to the pharmacodynamics, prognosis, and interactions with somatic mutations (*Bartsch et al., 2007*; *Erdem et al., 2012*; *Wang et al., 2018*; *Winther-Larsen et al., 2015*).

GGOs observed on computed tomography are described as hazy areas but preserved broncho-vascular markings (*Austin et al., 1996*; *Lee et al., 2014*). Advances in high resolution computed tomography and its application in lung cancer screening have led to an increased detection rate of GGOs, with an estimated prevalence of 0.2–0.5% (*Henschke et al., 2006*). While many GGOs are benign and disappear with time, some are persistent and turn malignant. These tumours are frequently found in non-smokers and women lung cancer patients (*Blons et al., 2006*; *Raz et al., 2006*).

Here, we recruited a total of 13 non-smoking patients with malignant GGO nodules to study their germline mutations using whole exome sequencing (WES). The results provide a better understanding of molecular mechanisms underlying the development of GGOs and their predisposition to turn cancerous.

## MATERIALS & METHODS

### Study subjects

Candidates that were radiologically found to have small GGO nodules in physical checkup or who came to outpatients department for the reason of cough and checked by computed tomography to have small GGO nodules were closely followed up from 6 months to 3 years. When the GGO nodules increased in size or the nodule density increased or the solid components of pulmonary nodules increased, 13 patients were recruited and underwent surgery and thereafter they were histologically confirmed to have malignant GGOs in the Department of Cardiothoracic Surgery at Wuxi People's Hospital affiliated to Nanjing Medical University, China, from April 1st, 2019 to August 30th, 2019. No other treatments were adopted. Written informed consent was obtained from all participants. Blood samples were collected before surgery and their clinical information was recorded. The research project was approved by the institutional review board of Wuxi People's Hospital affiliated to Nanjing Medical University (no: HS2019014).

### DNA extraction, library preparation, capture enrichment, and WES

Genomic DNA was extracted from peripheral blood collected from participants using a DNA blood mini kit (Qiagen, Germantown, MD, USA) following the manufacturer's instructions. DNA concentration and purity were assessed by a Qubit fluorometer (Invitrogen, Carlsbad, CA, USA).

**Table 1  Characteristics of study subjects.** Characteristics of study subjects was collected and analyzed.

| Characteristics | | |
| --- | --- | --- |
| Age at Diagnosis | Mean (SD) | 61.5 (8.7) |
| | Range | 48–79 |
| Gender | Male (%) | 2 (15.4) |
| | Female (%) | 11 (84.6) |
| Smoking history | Non-smokers (%) | 13 (100.0) |
| | Smokers (%) | 0 (0) |

WES was conducted on 500 ng of genomic DNA from each participant. Fragment libraries were prepared from sheared samples by sonication, and exons were enriched by hybridisation capture with a SureSelect Human All Exon V6 Kit (Agilent, Santa Clara, CA, USA) according to the manufacturer's protocol. Captured DNA was amplified followed by solid-phase bridge amplification. The paired-end library was sequenced on a NovaSeq 6000 platform (Illumina, San Diego, CA, USA). The data from this study were deposited in NCBI Sequence Read Archive under SRA accession: PRJNA613408.

**Read alignment, variant calling, variant annotation, and filtering**

Trimmomatic-0.36 (*Bolger, Lohse & Usadel, 2014*) was used as quality control for raw data and to remove adapters. Clean sequence reads were aligned to the human reference genome (GRCh37/b37 assembly) using Burrows-Wheeler Aligner software (version 0.7.10) (*Li & Durbin, 2009*). Picard (version 2.9.2, Broad Institute, Boston, MA, USA) was used to remove duplicates. Variant detection was performed using HaplotypeCaller in the Genome Analysis Toolkit 3.4 (https://gatk.broadinstitute.org/hc/en-us) (*DePristo et al., 2011*). Variants were annotated using InterVar database. Detailed stepwise filtering strategy for screening potential candidate germline mutations was described in Supplement 1.

**KEGG pathway analysis**

KEGG pathway analysis was conducted *via* KOBAS 3.0 (http://kobas.cbi.pku.edu.cn/).

## RESULTS

Clinical information of patients was summarised in Table 1. The mean age at onset of non-small cell lung cancer (NSCLC) in the cases was 61.5 years (range 48–79 years). All cases were non-smokers and 84.6% were females.

Two computed tomographic images are shown as representative of GGO nodules in the study cohort (Fig. 1). Of the 13 cases, 12 were diagnosed as lung adenocarcinomas while one was diagnosed as an atypical adenomatous hyperplasia. Five cases were adenocarcinoma *in situ*, four were invasive, and three were minimally invasive. Eight of the GGO nodules were located at the right upper lobe, two were at the right lower lobe, and three were at the left upper lobe. Detailed histologic information is presented in Table 2.

We used a stepwise filtering strategy to screen for potential candidate variants (Fig. 2). Of 83,302 single-nucleotide variants (SNVs) located in exons of the whole exome, our filtering strategy identified 17 potential candidate variants (Table 3). Of the 17 candidate variants

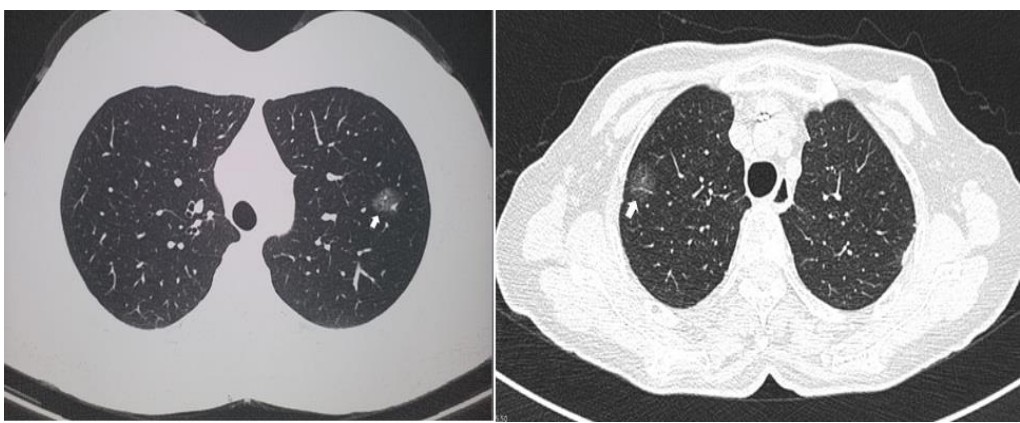

**Figure 1  Representative of ground glass opacity nodules.** Two representative computed tomography images of ground glass opacity nodules. The arrows indicate the nodules.

**Table 2  Clinical information of study subjects.** Pathology, tumor size and tumor location of study objects were shown in details.

| Patient ID | Pathology | Tumour size (cm) | Tumour location |
| --- | --- | --- | --- |
| WL-1 | LUAD[a], invasive | 1.2 | right upper lobe |
| WL-2 | LUAD[a], AIS[b] | 0.6 | left upper lobe |
| WL-3 | Atypical adenomatous hyperplasia | 0.5 | right upper lobe |
| WL-4 | LUAD[a], invasive | 1.5 | left upper lobe |
| WL-5 | LUAD[a], minimally invasive | 1.0 | right upper lobe |
| WL-6 | LUAD[a], minimally invasive | 0.6 | left upper lobe |
| WL-7 | LUAD[a], invasive | 2.0 | right upper lobe |
| WL-8 | LUAD[a], invasive | 2.0 | right lower lobe |
| WL-9 | LUAD[a], AIS[b] | 0.8 | right upper lobe |
| WL-10 | LUAD[a], AIS[b] | 0.7 | right upper lobe |
| WL-11 | LUAD[a], AIS[b] | 0.6 | right lower lobe |
| WL-12 | LUAD[a], AIS[b] | 0.8 | right upper lobe |
| WL-13 | LUAD[a], minimally invasive | 0.7 | right lower lobe |

**Notes.**
[a] lung adenocarcinoma.
[b] adenocarcinoma in situ.
[c] ground glass opacity.

in 17 genes, NM_000700:exon6:c.A418T in *AXAN1*, NM_001184790:exon8:c.C1070T in *PARD3*, NM_001170721:exon4:c.C392T in *BCAR1*, NM_001127221:exon46:c.G6587A in *CACNA1A*, NM_001170634:exon5:c.G383A in *FUS*, and NM_002451:exon6:c.C538T in *MTAP* were present in three cases each. In addition, rs756875895 frameshift in *MAX*, NM_001199292:exon7:c.C482G in *HSD17B4*, NM_005732:exon13:c.2165_2166insT in *RAD50*, NM_001350128:exon11:c.T1172C in *PPOX*, NM_001098816:exon28:c.A4751G in *TENM4*, NM_004004:exon2:c.235delC in *GJB2*, and NM_001142316:exon2:c.G203C in *LMO2* were present in two cases each. The remaining variants were present in one
A.

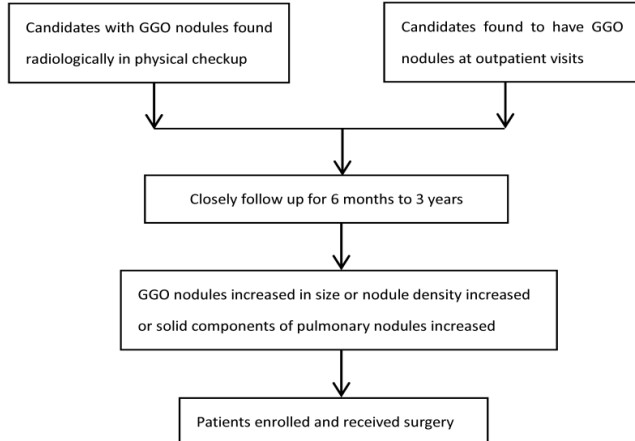

B.

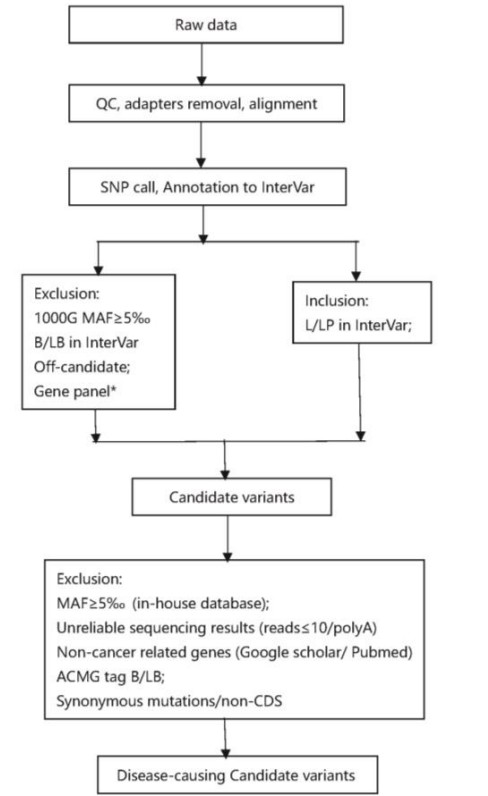

**Figure 2 Flowchart of analysis.** The stepwise filtering strategy used to screen for potential candidate germline mutations.

case each. Of the 17 variants, 12 were nonsynonymous mutations, four were frameshift deletions, and one was a stopgain. The distribution of mutations in each patient was shown in Fig. 3.

We identified four potential deleterious frameshift deletions: rs80338943 in *GJB2*, rs587781454 in *RAD50*, rs756875895 in *MAX*, each occurring in two cases, and a frameshift in *LDLRAP1*, occurring in one case. The first frameshift, rs80338943 in *GJB2*, causing a p.L79fs fusion, was annotated as uncertain significance by InterVar. The second frameshift, rs587781454 in RAD50 caused a p.K722fs fusion and was annotated as pathogenic from InterVar. The third frameshift, rs756875895 in *MAX*, was annotated as likely pathogenic in InterVar, causing a fusion of p.L52fs. The frameshift in *LDLRAP1* caused a fusion of p.W22fs in one case and was interpreted as pathogenic from InterVar, it might be deleterious in the development of lung cancer. Another potential deleterious variant was a stopgain, rs7755898 in *CYP21A2*, causing a protein change of p.Q289X which was likely pathogenic according to InterVar.

The other interesting candidates were four likely pathogenic SNVs annotated from InterVar: NOTCH 3:p.T357M (present in one case), HSD17B4 p.A161G (present in two cases), PPOX p.L391P (present in two cases), and TENM4 p.Q1584R (present in two cases).

There were five SNVs annotated as uncertain significance by InterVar that were present in three patients: ANXA1:p.I140 F, BCAR1:p.P131L, CACNA1A:p.R2196Q, FUS:p.S128N, and MTAP:p.R180W.

There were two additional candidate variants, LMO2 p.G68A and TTN p.R18629C, that were present in two cases and one case, respectively (Table 3). Their annotations by InterVar were of uncertain significance.

KEGG analysis did not indicate pathways that were related to *AXAN1*, *TENM4* and *GJB2*. Pathways of *BCAR1*, *CYP21A2*, *LPLRAP1*, *HSD17B4*, *MTAP*, *PPOX* and *TTN* were not associated with cancer. Pathways derived from *NOTCH3*, *PARD3*, *CACNA1A*, *MAX*, *RAD50*, *FUS* and *LMO2* were cancer-related. The details were shown in a (Table S). Mutations in these genes were considered unlikely to cause cancer, therefore they would not be discussed here.

## DISCUSSION

Although there are studies available on genetic mutations of lung cancer, the heritability of lung cancer, especially for GGO nodules, remains understudied compared to sporadic lung cancer. Using WES, our study reports germline mutations in GGO nodules of non-smoker lung cancer patients, largely females.

The discovery of germline mutations is very significant for both basic research and clinical treatment of lung cancer. First, germline mutations may play a role in tumorigenesis. *Wang et al. (2018)* reported that germline mutations interacted with somatic mutations, indicating their role in lung tumorigenesis. *Tomoshige et al. (2015)* also reported that germline mutations could cause familial lung cancer. Second, germline mutations are valuable for prognosis (*Erdem et al., 2012*). For example, a study by *Winther-Larsen et al. (2015)* found that genetic polymorphism in the epidermal growth

Mao et al. (2021), *PeerJ*, DOI 10.7717/peerj.12048

**Table 3  Summary of potentially deleterious germline mutations in lung cancer cases.** Annotation of potentially deleterious germline mutations in each gene were described in details.

| Gene | Position | RS | Ref/Alt | Protein alteration | Genetic model | Type of mutation | InterVar annotation | VAF in patients | VAF in GnomAD_EAS | No. of patients with mutation |
|---|---|---|---|---|---|---|---|---|---|---|
| ANXA1 | Chr 9: 75775752 | – | A/T | NM_000700:exon6:c.A418T:p.I140F | – | nonsynonymous SNV | Uncertain significance | 0.0833 | – | 3 |
| NOTCH3 | Chr 19: 15281611 | – | T/G | NM_001184790:exon8:c.C1070T:p.T357M | AD[a] nonsynonymous SNV | Likely pathogenic | 0.0278 | – | 1 |
| PARD3 | Chr 10: 34671665 | rs116642073 | G/A | NM_001184790:exon8:c.C1070T:p.T357M | – | nonsynonymous SNV | Uncertain significance | 0.0833 | 0 | 3 |
| BCAR1 | Chr 16: 75269775 | rs1047683608 | G/A | NM_001170721:exon4:c.C392T:p.P131L | – | nonsynonymous SNV | Uncertain significance | 0.0833 | 0 | 3 |
| CYP21A2 | Chr 6: 32008198 | rs7755898 | C/T | NM_001128590:exon7:c.C865T:p.Q289X | – | Stopgain | Likely pathogenic | 0.0278 | 0.0001 | 1 |
| LDLRAP1 | Chr 1: 25870253 | – | G/- | NM_015627:exon1:c.65delG:p.W22fs | AR[b] frameshift deletion | Pathogenic | 0.0278 | – | 1 |
| CACNA1A | Chr 19: 13319766 | rs373192655 | C/T | NM_001127221:exon46:c.G6587A:p.R2196Q | AD | nonsynonymous SNV | Uncertain significance | 0.0556 | 0.0015 | 3 |
| MAX | Chr 14: 65551007 | rs756875895 | G/- | NM_001271068:exon3:c.154delC:p.L52fs | AD | Frameshift deletion | Likely pathogenic | 0.0278 | – | 2 |
| HSD17B4 | Chr 5: 118814630 | rs763363391 | C/G | NM_001199292:exon7:c.C482G:p.A161G | AR | nonsynonymous SNV | Likely pathogenic | 0.0556 | 0.0002 | 2 |
| RAD50 | Chr 5: 131931460 | rs587781454 | -/T | NM_005732:exon13:c.2165_2166insT:p.K722fs | – | Frameshift deletion | Uncertain significance | 0.0556 | – | 2 |
| PPOX | Chr 1: 161140719 | – | T/C | NM_001350128:exon11:c.T1172C:p.L391P | AD | nonsynonymous SNV | Likely pathogenic | 0.0278 | – | 2 |
| FUS | Chr 16: 31195580 | – | G/A | NM_001170634:exon5:c.G383A:p.S128N | AD | nonsynonymous SNV | Uncertain significance | 0.0556 | – | 3 |
| MTAP | Chr 9: 21854717 | rs891972796 | C/T | NM_002451:exon6:c.C538T:p.R180W | AD | nonsynonymous SNV | Uncertain significance | 0.0556 | 0 | 3 |
| TENM4 | Chr 11: 78412907 | – | T/C | NM_001098816:exon28:c.A4751G:p.Q1584R | AD | nonsynonymous SNV | Likely pathogenic | 0.0833 | – | 2 |
| GJB2 | Chr 13: 20763485 | rs80338943 | G/- | NM_004004:exon2:c.235delC:p.L79fs | AD | Frameshift deletion | Uncertain significance | 0.0556 | – | 2 |
| TTN | Chr 2: 179427779 | rs192360370 | G/A | NM_003319:exon154:c.C55885T:p.R18629C | AR/AD | nonsynonymous SNV | Uncertain significance | 0.0278 | 0.0038 | 1 |
| LMO2 | Chr 11: 33886202 | – | C/G | NM_001142316:exon2:c.G203C:p.G68A | | nonsynonymous SNV | Uncertain significance | 0.0556 | – | 2 |

**Notes.**
[a] autosomal dominant.
[b] autosomal recessive.

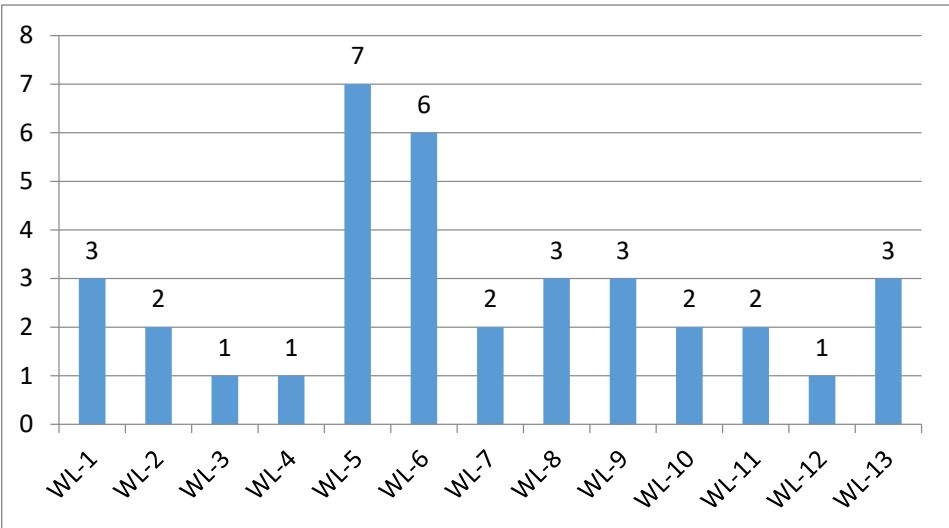

**Figure 3  Mutation distriubtion.** The distribution of germline mutations in each patient.

factor receptor could predict the outcome in advanced NSCLC patients treated with erlotinib. Third, germline mutations are closely associated with a genetic predisposition to cancer, and screening for germline mutations is beneficial to the susceptible population (*Chen et al., 2015*) and for their health management.

In this study, we used a highly selective population, lung adenocarcinoma patients with GGOs, to investigate germline mutations and their possible role in the predisposition to lung cancer. In our cohort, 11 of 13 were females and all were non-smokers. The ethnicity of all patients was Han Chinese. The aforementioned facts were consolidated with the notion that malignant GGO nodules occur frequently in non-smokers and women (*Blons et al., 2006*; *Raz et al., 2006*).

Strong evidence for two deleterious germline mutations (rs587781454 in *RAD50* and rs756875895 in *MAX*) has been shown in lung cancer patients. rs587781454 in *RAD50* was reported as a hereditary predisposition and labelled as pathogenic in ClinVar (*Nykamp et al., 2017*). rs756875895 in MAX was labelled as likely pathogenic by InterVar. Both variants occurred simultaneously in two females (WL-5 and WL-6). Both had minimally invasive GGO nodules. How these mutations in the same patient affected lung tumorigenesis is worth examining.

There was one likely pathogenic variant in *NOTCH 3* (WL-13). The expression of *NOTCH 3* was inversely associated with the sensitivity to platinum-based chemotherapy in patients with NSCLC. The NOTCH 3 protein, rather than the gene polymorphism, is associated with the chemotherapy response and prognosis of advanced NSCLC patients (*Shi et al., 2014*).

Though annotated as uncertain significance by InterVar, three patients carried variants in *BCAR1* (WL-7, WL-10 and WL-13) and *CACNA1A* (WL-5, WL-6 and WL-9). Increased expression of *BCAR1* was associated with poor prognosis and carcinogenesis

in NSCLC (*Deng et al., 2013*; *Huang et al., 2012*). Overexpression of *CACNA1A* predicted a poor prognosis in NSCLC (*Zhou et al., 2017*). There were one additional candidate variants, LMO2 p.G68A in WL-1 and WL-8. Collectively, these findings suggest that germline mutations may function by regulating gene expression and thereby affect cancer development and/or prognosis.

Our study has limitations. First, the sample size is small. In our study, only non-smoker patients with malignant GGOs were enrolled. Second, gene expression was not investigated. Finally, the identified germline mutations have not been validated. These limitations restrict conclusions about their causative effects on tumorigenesis and their roles as biomarkers for prognosis or for treatment response.

## CONCLUSIONS

In summary, our results demonstrate potentially deleterious germline mutations in GGO nodules in non-smoking lung adenocarcinoma patients. These findings significantly expand the spectrum of genetic variants that may affect the response to therapies and patient survival and possibly increase the risk of being germline mutation carriers. However, due to the small patient samples, our observations encourage further studies. In future, prospective studies, expanding enrolled patients and functional studies should be performed to better understand their causative roles in tumorigenesis and prognosis, and to better manage patients' health.

## ACKNOWLEDGEMENTS

We thank Editage for the language editing.

### Funding

This work was supported by grants from the Young Medical Key Talents Project in Jiangsu province (QNRC2016193), the Precision Medicine Project of Wuxi Municipal Commission of Health and Family Planning (J201805), and the Youth Scientific Research Project of Wuxi Municipal Health Commission (Q201951). The funders had no role in study design, data collection and analysis, decision to publish, or preparation of the manuscript.

### Grant Disclosures

The following grant information was disclosed by the authors:
Young Medical Key Talents Project in Jiangsu province: QNRC2016193.
Precision Medicine Project of Wuxi Municipal Commission of Health and Family Planning: J201805.
Youth Scientific Research Project of Wuxi Municipal Health Commission: Q201951.

### Competing Interests

The authors declare there are no competing interests.

## Author Contributions

- Wenjun Mao, Ruo Chen and Mingfeng Zheng conceived and designed the experiments, authored or reviewed drafts of the paper, and approved the final draft.
- Rongguo Lu, Shengfei Wang and Feng Liu performed the experiments, authored or reviewed drafts of the paper, collected specimens, and approved the final draft.
- Huizhu Song analyzed the data, authored or reviewed drafts of the paper, and approved the final draft.
- Dan You analyzed the data, prepared figures and/or tables, authored or reviewed drafts of the paper, and approved the final draft.
- Yijun He performed the experiments, authored or reviewed drafts of the paper, collected the clinical information, and approved the final draft.

## Human Ethics

The following information was supplied relating to ethical approvals (i.e., approving body and any reference numbers):

The Ethical Committee of Clinical New Technology and Medical Research Wuxi People's Hospital approves this research project (no: HS2019014).

## Data Availability

Data are available in the National Center for Biotechnology Information Sequence Read Archive: PRJNA613408.

## Supplemental Information

Supplemental information for this article can be found online at http://dx.doi.org/10.7717/peerj.12048#supplemental-information.

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
