# Peer review of "Germline mutation analyses of malignant ground glass opacity nodules in non-smoking lung adenocarcinoma patients"

_PeerJ, doi:10.7717/peerj.12048_

## Round 0.1 · original submission · Major Revisions

Please follow the indications given by the referees and address all issues raised thoroughly.

Reviewer 1 ·

Basic reporting

In their work, the researchers focused on a very specific patient sub-set of patients harboring malignant GGO and identified specific germ-line mutations which might eventually be associated with cancer of risk development.

The manuscript is generally well written, with sufficient background and good use of references. The structure is professionally managed and Raw data are included.

Experimental design

The research question is well defined and of clinical relevance in terms of early diagnosis of lung cancer and differential diagnosis between benign and malignant GGO findings.

However, the patient sample is very limited in number and potentially biased by the fact that all the patients were non-smokers. The authors should state if these patients were identified at casual radiologic findings or as part of organized screening programs.

Furthermore, the timing of blood sample collection would be relevant. Was the blood collected before, during or after treatment? Was it repeated after treatment to make sure that these are germinal findings and not mutations in cfDNA related to the presence of neoplastic cells? If possible, a repeated blood collection after treatment might reinforce the concept that the findings are germinal and not caused by tumor shredding.

Since the patient number is limited, the authors should state the treatment which was received by the patients and more robust outcome data for these patients. This could be evaluated in order to define any possible mutation with prognostic significance within our limited population.

Another bias is represented by the lack of a control group including either healthy individuals or subjects with non-malignant GGO as comparison. This addition might reinforce the concept that the detected mutations are actually potential biomarkers of malignant pulmonary findings.

Validity of the findings

The findings of the manuscript are interesting and encourage additional studies designed to use circulating biomarkers for cancer screening. However, I have the following observations, which should be addressed either by further analyses (as suggested in study design) or at least by addressing all the issues in the discussion:

1) As previously stated, the results involve a small patient sub-group, even more selected by the non-smoker status, which is often associated with a different mutational landscape compared to smokers' lung cancer. Hence the findings cannot be generalized.

2) As a control group is not present, the direct association between the findings and tumor development can only be implied, but cannot be demonstrated at the current stage.

3) As never-smoker lung cancer is molecularly different from current smoker lung cancer, the analysis on tumor samples of a comprehensive mutational panel or at least the most relevant activating mutations (EGFR, ALK, ROS1, BRAF) should be performed, and these findings associations with blood samples should be stated.

4) Due to the current study limitations, the researchers should stress the concept that no conclusions for prompt use in clinical practice can be drawn, while the observations might encourage further studies, including prospective studies in screening populations.

Reviewer 2 ·

Basic reporting

This study examined germline variants of 13 patients who had GGO in their lungs and received pulmonary surgery.
The following points should be revised.

1. The authors should clarify why these 13 patients received pulmonary surgery although their legions were small. Was the surgery done by physicians' decision or patients' strong will? The authors should show a consort diagram in which how the patients were properly selected.

2. A table or figure that explain how the detected variants were carried in each patient (mutually exclusive or overlapped in a subset of patients?).

3. The authors should show variant allele frequency for the detected variants in (non-cancerous) study population. Please explain "in-house database" in Table 1 in detail.

4. GGO (CT) and pathological images for all patients should be shown.

Experimental design

Method of patient selection is unclear since the authors do not show a consort diagram in which how the patients were properly selected.

Method of variants selection is unclear since the authors used an in-house database which are not explained in the text.

Validity of the findings

Authenticity of this study is lacked since the methods for patient and variant selection are unclear.
Overlap and/or mutually exclusiveness of detected variants are also unclear.

Additional comments

The authors should clarify the methods for patient and variant selection. Overlap and/or mutually exclusiveness of detected variants should be shown and discussed.

---

## Round 0.2 · Minor Revisions

The authors adequately addressed all issues raised by the referees, there are only minor points to be addressed.

Reviewer 1 ·

Basic reporting

I am generally satisfied with the revised version of the manuscript and the authors' answers to the referees. Globally, the quality of the manuscript is significantly improved compared to the initial version.

Experimental design

The authors improved the experimental design in a generally satisfactory way.

In lines 71-80 (study subjects), I suggest a global rephrasing, with particular reference to the sentence "When the GGO nodules increased in size or the nodule density increased or the solid components of pulmonary nodules increased, We recruited 13 patients were recruited who were histologically diagnosed and confirmed as having malignant GGOs", as it is not really clear whether the patients underwent surgery or not after observation of size increase. Apart from this part, the other sections of the manuscript are generally fluent and clear.

Validity of the findings

The findings are generally valid, although the small patient sample limits the robustness of the information. As suggested, the authors stressed the limitations of the study in the discussion.

However, I would suggest to reinforce this concept also in the conclusion, eventually with a sentence immediately preceding line 207 (" Our observations encourage further studies").

Additional comments

I am generally satisfied with the revised version of the manuscript, I suggest only minor revisions based on what reported in the previous sections.

---

## Round 0.3 · accepted · Accept

All issues raised by the referees have been addressed adequately.